# Green Foaming of Biologically Extracted Chitin Hydrogels Using Supercritical Carbon Dioxide for Scaffolding of Human Osteoblasts

**DOI:** 10.3390/polym16111569

**Published:** 2024-06-01

**Authors:** Mariana Quintana-Quirino, Adriana Hernández-Rangel, Phaedra Silva-Bermudez, Julieta García-López, Víctor Manuel Domínguez-Hernández, Victor Manuel Araujo Monsalvo, Miquel Gimeno, Keiko Shirai

**Affiliations:** 1Laboratory of Biopolymers and Pilot Plant of Bioprocessing of Agro-Industrial and Food By-Products, Biotechnology Department, Universidad Autónoma Metropolitana, Mexico City 09340, Mexico; mquintana@cua.uam.mx (M.Q.-Q.); adrih_rangel@hotmail.com (A.H.-R.); 2Tissue Engineering, Cellular Therapy and Regenerative Medicine Unit, Instituto Nacional de Rehabilitación “Luis Guillermo Ibarra Ibarra”, Mexico City 14389, Mexico; pssilva@inr.gob.mx (P.S.-B.); julietagarlo22@gmail.com (J.G.-L.); 3Biomechanics Laboratory, Instituto Nacional de Rehabilitación “Luis Guillermo Ibarra Ibarra”, Mexico City 14389, Mexico; vdominguez@inr.gob.mx (V.M.D.-H.); vicaraujom@yahoo.com.mx (V.M.A.M.); 4Food and Biotechnology Department, Chemistry Faculty, Universidad Nacional Autónoma de México, Mexico City 04510, Mexico; mgimeno@unam.mx

**Keywords:** methanogels, hydrogels, chitin, scaffold, supercritical carbon dioxide, osteoblast

## Abstract

Chitin is a structural polysaccharide abundant in the biosphere. Chitin possesses a highly ordered crystalline structure that makes its processing a challenge. In this study, chitin hydrogels and methanogels, prepared by dissolution in calcium chloride/methanol, were subjected to supercritical carbon dioxide (scCO_2_) to produce porous materials for use as scaffolds for osteoblasts. The control of the morphology, porosity, and physicochemical properties of the produced materials was performed according to the operational conditions, as well as the co-solvent addition. The dissolution of CO_2_ in methanol co-solvent improved the sorption of the compressed fluid into the hydrogel, rendering highly porous chitin scaffolds. The chitin crystallinity index significantly decreased after processing the hydrogel in supercritical conditions, with a significant effect on its swelling capacity. The use of scCO_2_ with methanol co-solvent resulted in chitin scaffolds with characteristics adequate to the adhesion and proliferation of osteoblasts.

## 1. Introduction

The global production of crustaceans by aquaculture or catch generates millions of tons of waste, consisting of heads and exoskeletons. The majority is still disposed of but, to some extent, biomass is valorized for chitin extraction and to obtain other added-value products as a waste-free biorefinery approach, which is consistent with the circular economy [1]. Chitin has several applications, including biomedical uses, owing to its biocompatibility and biodegradability. It also has other biological properties, causing the acceleration of wound healing and cartilage and bone regeneration [2]. Nevertheless, the strong intra- and intermolecular hydrogen bonding in chitin hampers solvent infiltration. This makes processing challenging due to chitin’s low solubility in water and in most common organic solvents [2,3,4,5]. Non-conventional solvents, such as *N*,*N*-dimethylacetamide with lithium chloride, are effective in dissolving chitin, but dimethylacetamide is a hazardous corrosive substance that can degrade chitin [2,6]. Another approach that circumvents the use of risky chemicals is the mixture of calcium chloride with methanol [7], which can cleave hydrogen bond interactions. This dissolves chitin and avoids molecular mass degradation. In general, the development of biodegradable and biocompatible polymeric scaffolds with adequate mechanical properties and porosity, as well as high swelling capacity and satisfactory degradation rates, are of particular interest for regenerative medicine [5,8]. In this regard, the use of supercritical fluids (SCFs) has been successful in producing porous materials from several polymers, including polylactic acid and poly-ε-caprolactone (PCL) [9]. Among the SCFs, supercritical carbon dioxide (scCO_2_) stands out due to its advantages in processing polymeric materials, which include its low cost, lack of toxicity, non-flammability, and the absence of solvent residues in the products, with accessible critical pressure (73.8 bar) and temperature (304 K) [8]. The absence of polarity for scCO_2_, which limits its interaction with hydrophilic matrices, can be overcome by the addition of polar porogenic co-solvents, like low-boiling alcohols or acetone. Hydrogels and methanogels can be foamed by using scCO_2_ and co-solvents in an SCF system [6,10]. Regarding the solvent system studied in this study (calcium chloride/methanol), structural changes have also been reported as calcium attacks the amide bond in chitin side chains, destroying the rigid crystalline regions [11]. Therefore, the hypothesis was that the combination of the nontoxic solvent system calcium chloride/methanol with scCO_2_ could produce structural changes that might enhance foaming, resulting in a non-permanent porous matrix with suitable properties for the culturing of human osteoblasts.

## 2. Materials and Methods

### 2.1. Biologically and Chemically Extracted Chitin from Litopeaneus vannamei Cephalothoraxes

Heads and shells of the shrimp species *Litopenaeus vannameii* were kindly supplied by Netmar (Mexico City, Mexico). The food waste was generated after the separation of the edible parts and it was shipped to the laboratory (−9 °C). Later, it was crushed in a meat grinder (Torrey 32-3) (Torrey, Querétaro, Mexico) and stored at −20 °C until use.

The shrimp waste was thawed and mixed with sucrose (10 wt/wt%) and we cultured *Lactobacillus brevis* for 24 h in Man Rogosa Sharpe broth at 30 °C (5 vol/wt%). The level of carbon source and starter were previously established by Cira et al. [12]. We placed 8 kg of this mixture into a 10 kg column reactor. The reactor was incubated for up to 120 h at 30 °C. The fermented solid was treated with HCl (0.25 N at 25 °C) and NaOH (0.25 N at 25 °C) for 1 h in each step. This was followed by aqueous neutralization; samples were taken and dried for further characterization [13].

### 2.2. Production of Porous Chitin Scaffolds Using scCO_2_

Chitin (1 g) was dissolved in methanol (MeOH; 8.75 mL) (Meyer, Mexico City, Mexico), water (16.25 mL), and calcium chloride (25 g) (J. T. Baker, Mexico City, Mexico) in a sealed flask. The mixture was kept under stirring conditions for 48 h at room temperature to dissolve the chitin. Then, the solution was poured into Petri dishes (55 mm in diameter) with a known weight (21.5 g) and placed in a bioclimatic chamber (Memmert HPP110, Schwabach, Germany) at 298 K with 90% relative humidity to form hydrogels. Excess water was removed, and the material was washed with distilled water [7]. 

Hydrogels were submerged in water or MeOH (15 mL) as indicated for 15 min. Hydrogels presented an initial humidity of 90.8 ± 1.3% and 63.9 ± 1.9% for the treatments with water and MeOH, respectively. The formation of the porous scaffolds was carried out, using water or methanol as co-solvents for scCO_2_, at temperatures of 298, 313, and 353 K and pressures of 175, 200, 250, and 300 bar. Hydrogel samples (5.4 ± 0.4 g) were placed inside a st-316 cylindrical porous cell (mesh 40), which was placed in a 400 mL st-316 cylindrical high-pressure reactor (Filher S.A de CV, Mexico City, Mexico). This was equipped with a high-pressure valve (Swagelok, Solon, OH, USA), external ceramic heating jacket, and two thermocouples, one for measuring the temperature inside the reactor and the other for measuring the temperature in the external jacket; both thermocouples were monitored by a temperature controller device. Then, CO_2_ was fed into the reactor using a high-pressure syringe pump (ISCO 260D, San Antonio, TX, USA). The experimental pressure and temperature conditions of each sample were adjusted by the heating controller and the ISCO pump. After 2 h, rapid depressurization was conducted for each condition by the full opening of the reactor valve. Products were removed from the reactor and placed at 195 K for 24 h in a Revco Ultrafreezer (Watertown, MA, USA) and, then, lyophilized (Labconco Freezone 2.5 plus, Kansas City, MI, USA) [6,10]. Samples were named S_A_ and S_M_ for processing with scCO_2_/water or scCO_2_/methanol mixtures, respectively. 

### 2.3. Characterization of Chitin Scaffolds

Changes in the porous scaffolds were assessed based on mass difference in an analytical balance (Ohaus Pionner, Parsippany, NJ, USA). The diameter of the samples was measured with a Vernier (Metromex, Mexico City, Mexico) and thickness was determined in a Mitutoyo Absolute (Mitutoyo Corporation, Kawasaki, Japan) device. The amount of CO_2_ absorbed by the hydrogels was determined based on mass loss difference. Porosity was determined by measuring the void fraction of the scaffolds in the different treatments and the void fraction of untreated chitin (control) using Equation (1): (1)ϕ=1−ε
where *ε* is equal to ω/ω_0;_ ω is the weight of the scaffolds of the different treatments; ω_0_ is the weight of untreated chitin; and *Φ* is the porosity [14]. Porosity measurements were performed in triplicate.

Pore size distributions were measured via scanning electron microscopy (SEM) on a microscope (JEOL JSM-5900 LV, Tokyo, Japan) using ImageJ software version 1.51j8 (National Institutes of Health, Bethesda, MD, USA). The samples were dried and then coated with gold before microscopic examination. Transverse samples were also fractured and treated similarly. Ten different micrographs from different samples of each different scaffold were used to determine the pore size distribution. Infrared spectroscopy (IR) spectra were recorded in an ATR FTIR spectrophotometer (Spectrum 100 FTIR, Perkin Elmer, Beaconsfield, UK). The crystallinity index (C_I_) was obtained from the ratio of the crystalline and amorphous peaks intensities. The apparent size of the chitin crystal (Dapp) in the porous scaffolds was obtained by X-ray diffraction in a diffractometer (Bruker D8 Advance, Karlsruhe, Germany) using Equation (2).
(2)$$D_{app}=k\lambda /\beta_{0}\cos\theta$$
where *k* is a constant at 0.9; *λ* (angstroms) is the incident radiation wavelength; *β*_0_ (rad) is the width at half the height of the crystalline peak; and *θ* is the scan angle of the (110) plane diffraction [15].

The degree of acetylation (DA) was determined in all the materials via proton nuclear magnetic resonance ^1^H NMR in a spectrometer (Bruker Advance II 300, Rheinstetten, Germany), performed using DCl/D_2_O with 3-(trimethylsilyl) propionic acid as internal reference. The percentage of swelling, water absorption, and erosion of the scaffolds was evaluated at 303 K for 14 days. Swelling capacities and weight loss gravimetric determinations for the scaffolds were conducted by the immersion of the scaffold samples into distilled water; swelling and degradation measurements were performed in triplicate. Samples were cut (1 cm × 1 cm approximately) and their initial dry weight was recorded (*W*_0_). Then, samples were immersed in an excess of distilled water. The weight of the swollen samples was measured every 24 h after removing the excess of surface water with filter paper [16]. The percentage of swelling capacity (*S*) at equilibrium (when swollen samples weight did not change between two measurements within 24 h of immersion in distilled water) was determined by Equation (3):(3)$$S(\%)=\left[(W-W_{0}\right)/W_{0}]\times 100$$
where *W* is the sample weight, and *W*_0_ is the initial dry weight. Mass loss (erosion) was calculated using Equation (4), considering the initial dry weight (*W*_0_) and the final dry weight (*W_f_*) of the samples:(4)Erosion(%)=[(W0−WfW0)]×100

Mechanical tests were performed using cuboid 1.0 × 1.0 × 0.1 cm^3^ samples on a universal testing machine (Instron Corporation, MA, USA) with a 1 kN load cell and a transverse spindle speed of 2 mm min^−1^. Four different samples of each scaffold were tested for the determination of the scaffold’s mechanical properties. A compression module was obtained as the tangent slope of the stress–strain curve [17]. The contact angles of water (10 µL distilled water drop) with chitin scaffolds were measured in triplicate at room temperature. Digital images of the water droplets were obtained in a microscope of horizontal light Qx3 Intel with an image processor (Intel Corporation, Santa Clara, CA, USA), and they were analyzed with the ImageJ 1.51j8 software (National Institutes of Health, Bethesda, MD, USA). 

### 2.4. Characterization of the Osteoblasts Response in the Chitin Porous Scaffolds

Osteoblasts (hFOB 1.19, CRL-11372™ ATCC^®^ Manassas, VA, USA) were grown in culture flasks with a Dulbecco’s Modified Eagle Medium and Ham’s F12 Medium (DMEM/F12; Gibco, Thermo Fisher Scientific, Waltham, MA, USA) supplemented with 10% *v/v* fetal bovine (FBS; Gibco, Thermo Fisher Scientific, Waltham, MA, USA) and 1% *v/v* antibiotic-antimycotic (Anti-Anti; Gibco, Thermo Fisher Scientific, Waltham, MA, USA. Osteoblasts were incubated in a humid environment of 5% CO_2_ at 310 K, and the culture medium was replaced every 2 days until reaching 80% cell confluence. Cell cultures were washed and treated with 0.05% trypsin-EDTA solution to release the cells attached to the culture flask. Subsequently, cells were centrifuged for 10 min at 2500 rpm. Then, the cell pellet was resuspended in DMEM:F12 complemented with FBS and antibiotic–antimycotic treatment, and then seeded in culture flasks for further expansion of the number of cells. Cells were not expanded by this method more than three times (passes) before they were used in the actual experiments with the scaffolds.

#### 2.4.1. Cell Proliferation

Osteoblast proliferation in the scaffolds was determined via the reduction of bromide of 3-(4,5-dimethylthiazol-2-yl)-2,5-diphenyltetrazole (MTT assay). For this purpose, scaffolds were sterilized in an autoclave at 394 K and 15 lb in^−2^ for 15 min and cut into circular samples of 8 mm in diameter and 1 mm in thickness. Then, the samples were individually placed in a 48-well microplate and hydrated with Phosphate Buffer Saline solution (PBS 1X; Gibco, Thermo Fisher Scientific, Waltham, MA, USA). Osteoblasts were seeded in the scaffold’s samples at a density of 50,000 cells per scaffold. For positive controls, cells seeded in culture well plates (2000 cells cm^−2^) were used. At specific cell culture times (2, 4, 7, 14 and 21 days), independent cell-seeded scaffolds samples were individually incubated for 3 h with 15 µL of MTT solution (0.5 mg mL^−1^) and 150 µL of DMEM:F12. Subsequently, the MTT solution was removed and 150 µL of DMSO/isopropanol (1:1) solution was added for 10 min to dissolve the formazan crystals that were formed due to metabolization of MTT by cells. Finally, the absorbance of the formazan dissolution was measured at 570 nm in a microplate reader (Filter Max F5 Multimode Microplate reader, San Jose, CA, USA) [18]. The experiments were carried out in triplicate in two independent sets of experiments.

#### 2.4.2. Cell Morphology

The morphology, distribution, and presence of human osteoblasts cultured in chitin porous scaffolds was characterized by scanning electron microscopy (SEM) (JEOL JSM-5900 LV, Tokyo, Japan) and hematoxylin and eosin (H&E) staining on independent scaffolds samples. For SEM evaluation, after 21 days of cell culturing on the scaffolds, the cell-seeded scaffolds samples (constructs) were washed with PBS and fixed in 2% formaldehyde for 4 h. Then, the samples were washed with PBS again and dehydrated twice based on the gradient of the ethanol solution (30, 40, 50, 70, 80, 90, and 100%) for 15 min. After the final dehydration step with 100% ethanol, the samples were dried with CO_2_ at the critical point, fixed with colloidal carbon, and coated with gold prior to SEM analysis. For H&E evaluation, independent cell-seeded scaffolds samples, maintained in culture for 15 days, were washed with distilled water and fixed with absolute ethanol for 1 min. Then, the samples were washed with distilled water twice and stained for the visualization of nuclei in blue purple with Harris hematoxylin/eosin. Micrographs were acquired in an Axio Imager Z2, Carl Zeiss microscope at 5X and 10X magnifications. Experiments were carried out in duplicate. 

#### 2.4.3. Cell Viability

The viability of human osteoblasts upon culturing in the scaffolds was directly assessed via the LIVE/DEAD^TM^ Viability/Cytotoxicity Kit for mammalian cells (Invitrogen^®^, Waltham, MA, USA) performed at 15 days of cell culturing in the scaffolds. After the culture time interval, the scaffolds were washed twice with PBS and cut across into 1 mm thick slices to evaluate the cell viability, but also cell penetration into the scaffold. Then, the LIVE/DEAD assay (LIVE/DEAD™ Viability/Cytotoxicity Kit, for mammalian cells; Invitrogen^®^ Waltham, MA, USA) was performed on the samples’ slices according to the instructions from the kit manufacturer. Finally, samples were rinsed twice with PBS and immediately visualized via fluorescence microscopy (Axio Imager Z2, Carl Zeiss, Jena, Germany). Images were processed using the AxionVision software^®^ (Version Rel 4.8.2, Carl Zeiss, Jena, Germany). Experiments were carried out in triplicate.

#### 2.4.4. Cell Functionality

The functionality of the osteoblasts upon culturing with the scaffolds was evaluated by immunofluorescence (IF) assays against osteocalcin as a characteristic protein, expressed by osteoblastic phenotype cells. After 15 days of culture, independent cells–scaffold samples were rinsed twice with PBS and fixed with paraformaldehyde (PFA, 2%, Sigma Aldrich, Waltham, MA, USA) at 4 °C. Fixed samples were permeabilized with 0.25% triton in PBS (PBST) and blocked with 1% albumin and glycine 22.52 mg/mL in PBST. Independent samples were incubated at 4 °C with the addition of the primary antibody to osteocalcin (Abcam ab 93876; rabbit primary anti-human antibody, 1:100). Then, the primary antibody was removed, and samples were incubated at room temperature with the corresponding secondary fluorescent antibody from Alexa Fluor 488^®^ (1:100; donkey anti-rabbit, Abcam ab 150073). Samples were washed with PBS, and examined by fluorescence microscopy (Axio Imager Z2, Carl Zeiss, Jena, Germany); cell nuclei were counterstained with Hoechst (33342 Invitrogen). Experiments were carried out in duplicate. 

### 2.5. Statistical Analysis

NCSS program version 7.0 (NCSS Inc., East Kaysville, UT, USA) was used to determine the significance among the different scaffolds studied. The means of the results were compared with Tukey–Kramer multiple means comparison test (*p* ≤ 0.05). 

## 3. Results and Discussion

### 3.1. Sorption of scCO_2_ on Porous Chitin Scaffolds 

Chitin hydrogels for the SCF treatment had an average mass of 5.4 ± 0.4 g, a thickness of 1.8 ± 0.2 mm, and a diameter of 5.9 ± 0.3 cm. The effect of the co-solvent on the formation of porous scaffolds from the hydrogels showed significant differences in the mass and diameters of the products (Appendix A). The mass loss curves for the S_A_ and S_M_ scaffolds due to the desorption of CO_2_ at room temperature in Figure 1A,B displayed an initial linearity (upper red trace) for the combined desorption of CO_2_ and loss of moisture. However, after 300 s (lower red trace), the loss of mass only corresponded to that of moisture. The linear regression of both sections provided the mass of the absorbed CO_2_; shrinkage and significant mass loss were observed after the foaming process with scCO_2_, using water and methanol as co-solvents. Similarly, Tsioptsias and Panayiotou [6] reported this behavior for porous chitin materials. 

Of note, Figure 1C,D evidenced higher mass loss in the scaffolds treated with methanol than in those treated with water. Similarly, Tsioptsias et al. [10] indicated that the sorption of CO_2_ in hydrogels was favored when water was replaced by methanol in the production of porous chitin materials from hydrogels. Concomitantly, after foaming, the materials presented a loss of mass and shrinkage according to the weight and diameter measurements.

Table 1 shows the amount of CO2 absorbed during the foaming process at several temperatures and pressures. The experimental evidence showed the highest absorption to be at 353 K and 175 bar.

### 3.2. Chemical Characterization of Scaffolds

The characteristic FTIR bands for the stretching of the C-O single-bond vibrations of the carbonyl groups were observed at 1085, 1125, and 1160 cm^−1^. The C=O stretching for the amide I in α-chitin appeared as a doublet between 1620 and 1660 cm^−1^, and the bands at 1560 cm^−1^ and 1562 cm^−1^ were assigned to the NH flexion of the amide II for α-chitin. The asymmetric and symmetrical stretching of the NH functional group appeared at 3250 and 3167 cm^−1^, respectively. Other bands were also assigned to the stretching of the O-H groups at 3460 cm^−1^ and to that for C-H at 2877 cm^−1^. The other characteristic band for this polysaccharide owing to the glycosidic bond was observed at 895 cm^−1^ [13,14,15,16,17,18,19]. Other bands for the C-O were found at 1068 and 1153 cm^−1^. We also observed amide I at 1620 cm^−1^, the flexion of the amide II at 1556 cm^−1^, and the asymmetrical stretching of the NH group at 3258 cm^−1^ (Appendix A shows the FTIR spectra of the samples at different conditions). 

On the other hand, the characteristic peaks in the chitin X-ray diffraction pattern, as reported elsewhere [19], were also observed for the pure chitin in the present study (Figure 2). Strong reflection peaks were observed at 2θ = 9–10° and 20–21° for the (020) and (110) diffraction planes, respectively. Conversely, peaks with smaller intensities, corresponding to the (101) and (130) diffraction planes, were observed at 2θ = 21–26°. The chitin used in the present work had a C_I_ of 85.5 ± 0.4%, a D_app_ of 7.6 ± 0.2 nm, and a DA of 99.8 ± 0.5%. Figure 2 shows the X-ray diffraction patterns of the chitin hydrogels after scCO_2_ treatment using water or methanol at 298 K and 175 bar (Figure 2A). The analyses of the spectra evidenced that both materials kept the peak of highest chitin reflection (2θ = 19–20°) from the native chitin. However, the second characteristic peak at 2θ = 9–10° was less intense for the S_M_, which might suggest a reduction in the crystallinity of the material when foaming with scCO_2_/MeOH. The use of methanol as a co-solvent facilitated the solubility of CO_2_ in chitin, favoring the absorption and interaction of this inorganic fluid with the polymer molecules by breaking intramolecular chain bonding, and thereby modifying the crystalline arrangement. 

The X-ray diffraction spectra for the materials, obtained at several temperatures and pressures, are presented. Figure 2B shows the expected characteristic peak (2θ = 19–20°) for all samples. However, the lowest reflection (2θ = 9–10°) peak decreases as the temperature increases. This temperature-dependent behavior followed a well-known pattern for semi-crystalline polymers, as increasing mobility of the chains allows for the interpenetration of the SCF, thus disrupting the crystalline arrangement. However, the opposite effect was observed when increasing the pressure. Therefore, the increase in density with pressure, which usually enhances the interpenetration of scCO_2_ in solid matrices, cannot be reproduced for these chitin hydrogels. The latter effect might be ascribed to the decreasing gap between chitin chains as the pressure increases, which might favor the strengthening of the hydrogen bonding and hence increase crystallinity, thus limiting SCF diffusion [20,21].

Table 2 shows the C_I_, D_app,_ and DA of chitin after SCF foaming with significant changes compared to untreated chitin samples. The S_M_ material obtained at 353 K and 175 bar showed a greater deacetylation than initial chitin, with a DA reduction between 9 and 22%. 

### 3.3. Effect of Co-Solvent Morphology and Porosity of Samples

SEM micrographs of S_A_ showed a rough surface, with the presence of non-homogenous pores (Appendix A). Additionally, the formation of interchannels was observed, which was most sought after for cell proliferation (Appendix A). Meanwhile, S_M_ displayed an even rougher surface than that of SA and the porous distribution on the surface was enhanced (Appendix A), which was corroborated by porosity (S_M_ = 62 ± 0.7% and S_A_ = 21 ± 7%) and pore distribution (Figure 3C) determinations. The pores also became evident in the internal part of the latter material, showing the formation of interconnectivity (Appendix A).

Porosity and pore diameter, which are important for tissue engineering as well as for suitable liquid absorption, were measured by pore counting in the SEM micrographs using the ImageJ software (1.51j8 version). The results displayed 1.58-fold more pores for S_M_ compared to S_A_ sample (Figure 3A). S_M_ had a porosity of 62 ± 0.71%, with an average pore diameter of 1.0 ± 0.9 µm, which was 3-fold higher than that of S_A,_ with a porosity of 21 ± 7.0% and an average pore diameter of 1.4 ± 2.4 µm (Figure 3B). This agrees with a report by Diaz-Gómez et al. [9] showing similar porosities, 62–64%, in hybrid scaffolds of PCL–starch. These authors also reported a good distribution of the pore diameter for pores formed by scCO_2_-mediated treatment of hydrogels. It is worth noting that the S_M_ scaffold had a heterogeneous pore distribution, as shown in Figure 3C, unlike the SA scaffold, having pore diameters between 0.5 and 1.0 µm (Figure 3C). On the other hand, in Figure 3D,E, it can be observed that S_M_ presented greater water absorption (3.0 ± 0.3 g of water per g of the scaffold material) and a higher mass loss (≈22% at 12 days of immersion in distilled water) upon immersion in distilled water than S_A_ (swelling of 1.3 ± 0.1 g of water per g of the scaffold material and ≈8% mass loss at 12 days of immersion in distilled water), which was attributed to the specific porosity characteristics of the two different scaffolds studied. 

### 3.4. Effect of Temperature on Morphology and Porosity for S_M_ Sample

The morphology in the S_M_ scaffold, treated at constant pressure (175 bar) but varying temperatures, showed a rough surface, with the formation of multiple pores of different diameter for all samples. However, the smallest pores on the surface were observed at 298 K (Figure 4A (298-S and T)). Similarly, the internal area of the material presented an average pore diameter of 1.7 ± 0.04 µm, which was significantly larger than that attained for the other samples (53 ± 4.4%). This is consistent with the results described by Ye et al. [20] for PCL and poly (ethylene oxide) materials, where the formation of pores with small diameters was favored at room temperature. Moghandam et al. [22] described that below 318 K, PCL did not fully fluidize; in turn, the SCF solubility in the polymer was low, resulting in inappropriate porosity. However, above 318 K, the polymer was completely fluidized, thereby increasing the interpenetration of CO_2_ and consequently the porosity, albeit with a non-uniform distribution of the pore diameters. PCL characteristics differ from those of chitin; notwithstanding, the higher the temperature, the higher the SCF interpenetration that also occurs in chitin hydrogels (as previously discussed for the X-ray diffraction pattern). This, in turn, improves porous formation in the polysaccharide. In agreement with this, formations of pores of various diameters were distributed on the surface in a cross-sectional area of S_M_ scaffolds treated at 313 K, which showed an average pore diameter of 2.0 ± 0.1 µm and a porosity of 33 ± 2.6% (Figure 4A (313-S, T)). However, at 353 K, despite the similar morphology to the previous treatments (Figure 4A (353-S, T)) with a porosity of 59 ± 3.1% and an average pore diameter of 2.8 ± 0.04 µm, the pores on the surface and in the internal zone were uniform and interconnected. Ye et al. [20] observed a similar behavior when producing porous matrices of PCL/poly (ethylene oxide) mixtures, and they explained that materials with a larger pore size were obtained at temperatures higher than 298 K. The results for chitin hydrogels were also analogous to those reported by Tai et al. [21] for the production of poly(*D*,*L*-lactic acid-co-glycolic acid) scaffolds because the increase in temperature (<328 K) produced large and open pores, which was attributed to the greater diffusion of CO_2_ into the polymer.

SEM imaging processing, which is graphically represented in Figure 4B, provided further evidence of the temperature-dependent average pore diameter in samples. The distribution of the pore diameters was mostly dispersed for the three temperature treatments (Figure 4C). However, most of the pores produced at 298 K had a diameter between 0.1 and 0.5 µm, whereas at 313 and 353 K, there was uniformity, centered at 0.5 and 1.0 µm (Figure 4C). Interestingly, the water absorption capacity depended on the porosity characteristics, so there was significantly more absorption (3.0 ± 0.1 g water g scaffold^−1^) at 353 K than in the samples from the others treatments, as can be seen in Table 1.

### 3.5. Effect of Pressure on Morphology and Porosity of S_M_ Scaffolds at 353 K

The S_M_ scaffolds obtained at different pressures and constant temperatures of 353 K presented rough surfaces (Figure 5A) with uniform pore formation. The samples obtained with 200, 250, and 300 bar had small-diameter pores (Figure 5A (S: 200, 250, 300)). On the contrary, the hydrogels processed at 175 bar presented relatively large superficial pores with uniform diameters (Figure 5A (175-S)), which were also highly interconnected (Figure 5A (175-T)). On the other hand, closed-type pores with larger diameters and sections with heterogeneous pores were observed for the other treatments.

The average pore diameter data, as shown in Figure 5B, evidenced that the pore diameter decreased when the pressure increased. This was consistent with the work of Moghamad et al. [22] to produce PCL scaffolds. The authors described that this is related to the decreasing viscosity of PCL, which restricts the interaction between the CO_2_ and the polymer. Ye et al. [20] reported a similar behavior for PCL/poly (ethylene oxide) materials when the pressure increased above 15 MPa. This was attributed to the decreasing of the melting temperature of the polymer mixture, which facilitated CO_2_ absorption. Figure 5C shows that the distribution of the pore diameter at the highest pressure was between 0.1 and 0.5 µm, while at 175 bar, the average pore diameter was between 1.0 and 5.0 µm. In the present study, the variations of the porosities in the materials with operational pressures were 62 ± 5.3% for 175 bar, 44 ± 1.9% for 200 and 250 bar, and 47 ± 9.8% for 300 bar. On the other hand, the highest absorption was obtained for S_M_ samples processed at 175 bar (3.5 ± 0.7 g water g scaffold^−1^), which was 1.6-fold higher than other treatments (Table 1). Therefore, the S_M_-derived porous material obtained at 353 K and 175 bar displayed the most promising characteristics as potential scaffolds for use in further biological studies [9]. 

### 3.6. Swelling, Erosion and Contact Angle Determination

The potential scaffolds obtained presented contact angles similar to those of native chitin. The latter showed a water drop with a spherical shape (θ = 100 ± 3.1°) that even exhibited hydrophobic characteristics in the form of hydrogel (Appendix A). However, for S_M_, which had higher porosity than S_A_, the analysis displayed a significantly low contact angle (Table 2), which was explained by the rapid absorption of the drop on the surface (Figure 3A). The different temperature treatments also presented significant differences in the contact angle, with respect to the control (Appendix A), ranging from the initial 100 ± 3.1° to a remarkable decrease in contact angle with treatment temperature for S_M_ samples (Table 3). Similarly, pressure changes also displayed significant variations among samples (Table 3): the higher the porosity, the lower the contact angle (Appendix A).

The results of the swelling and erosion assays, shown in Figure 3D,E, corroborated these differences, as S_M_ presented a higher percentage of swelling and erosion than S_A_. This evidence agreed with Ji et al. [17] regarding chitosan scaffolds from hydrogels cross-linked with glutaraldehyde and genipine, where the materials with less swelling were also those with the lowest porosity.

The maximum swelling and erosion percentages were consistent with the results of the contact angle for S_M_ (Table 3). Although the increase in swelling was related to porosity, the erosion presented by S_M_ scaffolds was attributed to the crystalline structure obtained after foaming with scCO_2_. In this regard, Villa-Lerma et al. [23] reported the modification of the crystalline structure in the processing of chitin by supercritical 1,1,1,2-tetrafluoroethane, as the reduction of C_I_ was also observed.

### 3.7. Mechanical Properties

The co-solvent used in the present work played an important role in the mechanical properties of the materials, as shown in the compression profiles of S_A_ and S_M_ (Figure 6A). It is worth noting that S_M_ presented the lowest compression, which was attributed to the modification of its internal structure, favoring fragmentation with a lower elastic modulus of 14.4 ± 1.4 MPa in comparison with S_A_ (37.9 ± 8.2 MPa). Ji et al. [17] stated that the compression module depends on the nature of the material and its pore size. In this regard, Figure 6B displays the stress–strain profile of the porous materials processed at different temperatures, where a proportional effect on the compressive strength can be observed. According to this information, the porous materials suffered less deformation when the compression resistance was low, meaning that there was therefore a high Young’s modulus. Table 3 shows the Young’s moduli of the different materials in terms of what was reached for each operational temperature. Interestingly, the S_M_ scaffold at 353 K displayed the smallest module, while a high modulus was displayed at 298 K. This was attributed to the different porosities of the materials. This was consistent with the result of Ji et al. [17] when producing chitosan scaffolds cross-linked with glutaraldehyde and genipine. These authors observed that the materials had a low elastic modulus, related to the porosity achieved in their scCO_2_ foaming process. On the other hand, in the present work, the pressure also influenced the mechanical strength of S_M_ samples, as the lowest deformation was encountered at the highest porosity (Figure 6C). The S_M_ material from the treatment at 175 bar presented a significantly smaller Young’s module (Table 3) than the other treatments, which was substantiated by the porosity determinations. It is worth mentioning the previous studies on porous materials for cell proliferation in tissue engineering studies, such as human skin fibroblasts [22] and human mesenchymal stem cells [17,18]. In general, cells require suitable mechanical properties to prevent deformation under physiological loads with an elastic modulus higher than 10 MPa [22]. This datum substantiates the potential use of the porous supports produced in the present study as scaffolds for regenerative medicine [24].

### 3.8. Cell Culture of Human Osteoblasts in the S_M_ and S_A_ Scaffolds

The proliferation of human osteoblasts in the scaffolds was assessed by the MTT assay for the S_M_ and S_A_ samples, obtained at the best processing conditions with regard to pressure and temperature (175 bar and 353 K). After 2, 7, and 14 days of incubation, the number of cells showed no significant differences for either scaffold; however, cell proliferation was observed in both scaffolds. The level detected was double the number of cells in the S_M_ and the S_A_ scaffolds from day 2 to day 14 of cell culturing. After day 14 of cell culturing, the cell proliferation rate in S_A_ decreased in comparison with that in S_M_. By day 21, the S_M_ scaffold showed the highest cell proliferation, with a significantly larger number of cells compared to S_A_, where the number of metabolically active cells did not significantly increase from day 14 to day 21 of cell culturing. In S_M_, the number of metabolically active cells increased two-fold at 21 days of culture in comparison with the number of cells observed in the same scaffolds at 14 days of cell culture (Figure 7).

The presented cell proliferation results are similar to those reported by Moghamad et al. [22] using porous scaffolds of high-molecular-weight PCL and hydroxyapatite for the proliferation of human mesenchymal stem cells from bone marrow. These authors reported an increase in the number of metabolically active cells over time owing to the biocompatibility of the scaffold, which led to the conclusion that the use of porous supports promotes proliferation and cell growth. On the other hand, Liu et al. [24] observed enhanced osteoblast (MC3T3-E1) growth over time in chitin nanocrystals and chitosan composite scaffolds, and this was ascribed to the high porosity of the material used for scaffolding, which in turn allowed appropriate cell movement and nutrient flow for cell proliferation. 

The viability of the human osteoblasts cultured in the S_M_ and S_A_ scaffolds was directly evaluated at 14 days of cell culture by the calcein AM/ethidium homodimer assay (Figure 8A). In both scaffolds, viable cells (green) were observed, forming monolayers on the surface of the scaffolds. On the other hand, the side view of the scaffolds showed a greater penetration of the cells into the S_M_ scaffold, in comparison to the cell penetration observed in the S_A_ scaffold. In the same sense, the number of viable cells seemed to be higher in the S_M_ scaffold (forming larger cells monolayers) in comparison with the S_A_ scaffold, and dead cells (red) seemed to be fewer in the S_M_ scaffold in comparison with the S_A_ scaffold. This was in agreement with the MTT results. These showed that there were viable cells in both scaffolds, but that there was a trend of cell proliferation and number of viable cells where the S_M_ scaffold seemed to be more biocompatible than the S_A_ scaffold in terms of supporting osteoblast culture. Cell functionality in the scaffolds was assessed by the expression of osteocalcin by immunofluorescence assays (Appendix A). A slightly larger fluorescence intensity was observed from the scaffolds incubated with osteoblasts, in comparison with the Ctrl- (scaffolds incubated with no cells); nevertheless, auto-fluorescence from chitin scaffolds prevented the clear identification of osteocalcin expression from osteoblasts cultured in the scaffolds.

The morphology of the osteoblasts upon 21 days of culture in the S_M_ and S_A_ scaffolds, as evaluated by SEM, is shown in Figure 8B,C. Osteoblasts present on the surface of the scaffolds displayed the characteristic morphology of well-adhered cells for both scaffolds. Visibly, a greater number of cells was observed to adhere to S_M_ throughout enhanced production of cellular matrix, suggesting better cytocompatibility, growth stimulation, and cell viability. SEM results were corroborated by the H&E assay (Figure 8D,E) where, at 14 days of cell culture, monolayers of cells were observed in both scaffolds; however, cell monolayers on the S_M_ scaffolds covered larger areas of the scaffold in comparison to S_A_. Through H&E staining, it is normally possible to observe the matrix and the nuclei of the cells. Nevertheless, in the present case, the scaffolds presented an unspecific weak staining that prevented the clear identification of the extracellular matrix; however, the cell nuclei were undoubtedly distinguished. Finally, it is important to emphasize that the SEM and H&E results corroborated, in addition to supporting the cell viability and proliferation results, that the use of chitin hydrogels mediated by SCF technology using CO_2_ allows the non-toxic manufacture of porous chitin scaffolds, with appropriate properties for use in cell culture.

## 4. Conclusions

The use of scCO_2_ technology allowed the efficient production of chitin scaffolds by the rapid desorption of this inorganic solvent, where the use of methanol co-solvent had a favorable effect on the formation of chitin pores. The effect of pressure and temperature on the formation of pores in the materials was also assessed. This showed that the best operational conditions were 175 bar and 313 K and that these were obtained using a methanol/scCO_2_ solvent. The materials produced at these pressure and temperature conditions were the best in terms of porosity, average pore diameter, and swelling as potential cellular scaffolds. They were biocompatible, promoting cell adhesion and consequently allowing the appropriate culture and proliferation of human osteoblasts on them. The S_M_ scaffold showed the best properties for cell culture and demonstrated potential applications for bone tissue engineering.

## Figures and Tables

**Figure 1 polymers-16-01569-f001:**
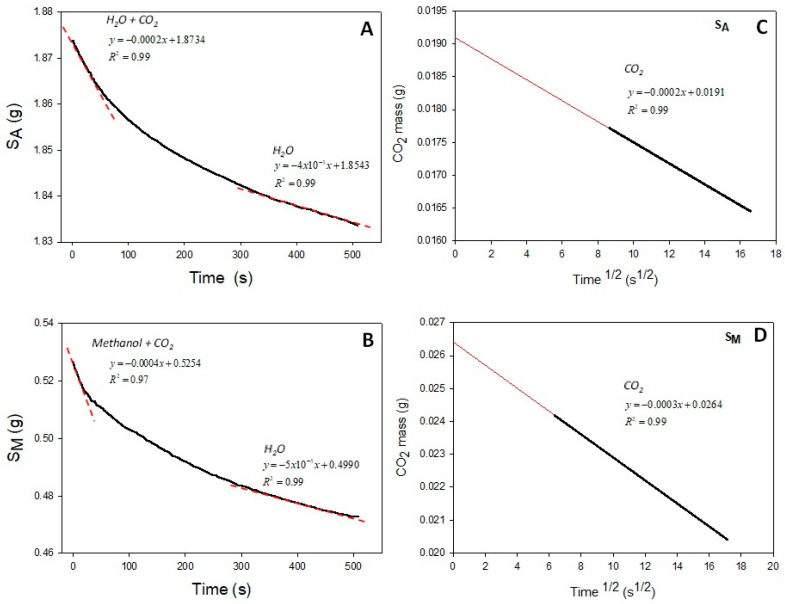
Representative desorption curves for S_A_ and S_M_ (**A**,**B**) and mass loss of both materials due to CO_2_ desorption (**C**,**D**) at 353 K and 175 bar. The dotted lines represent the fitted data.

**Figure 2 polymers-16-01569-f002:**
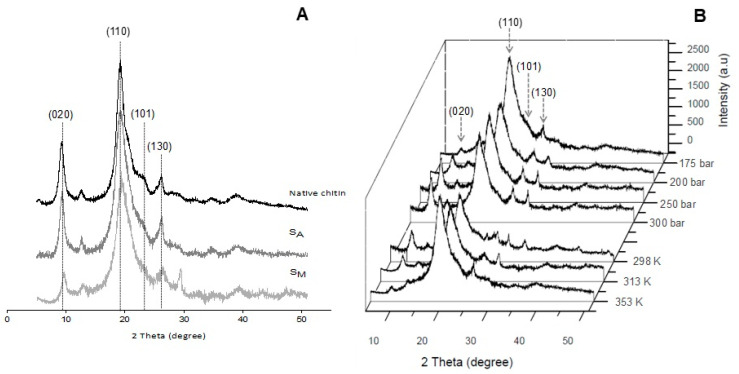
X-ray diffraction patterns of native chitin and chitin treated with scCO_2_ (**A**) at 175 bar and 353 K using water (S_A_) and methanol (S_M_) as co-solvents, and (**B**) S_M_ scaffolds obtained at different temperatures and pressures.

**Figure 3 polymers-16-01569-f003:**
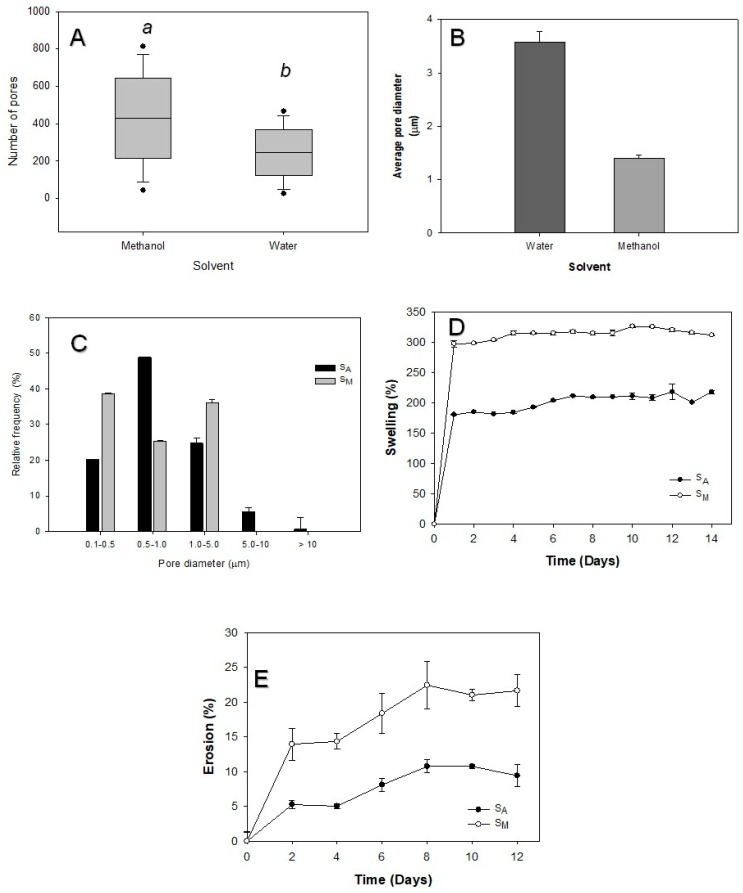
Number of pores (**A**), average diameter (**B**), pore distribution (**C**), swelling (**D**) and erosion (**E**) for SCF-treated S_A_ and S_M_ samples at 353 K and 175 bar. Different letters mean statistically different results (Tukey–Kramer *p* ≤ 0.05).

**Figure 4 polymers-16-01569-f004:**
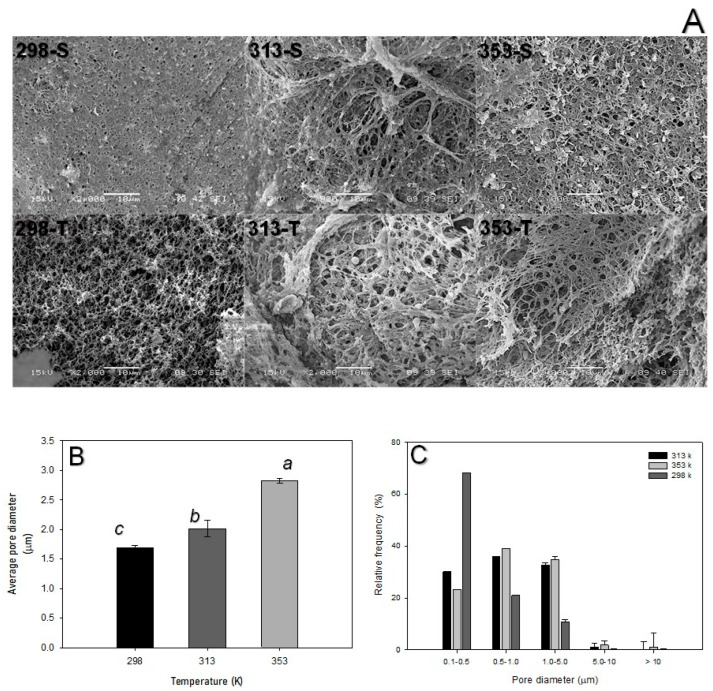
SEM micrographs of S_M_ after scCO_2_ (S: surface, T: transverse) (**A**). Average diameter (**B**) and distribution of the pores (**C**) at different temperatures and constant pressure (175 bar). The different letters in the histograms mean that they are statistically different (Tukey–Kramer *p* ≤ 0.05).

**Figure 5 polymers-16-01569-f005:**
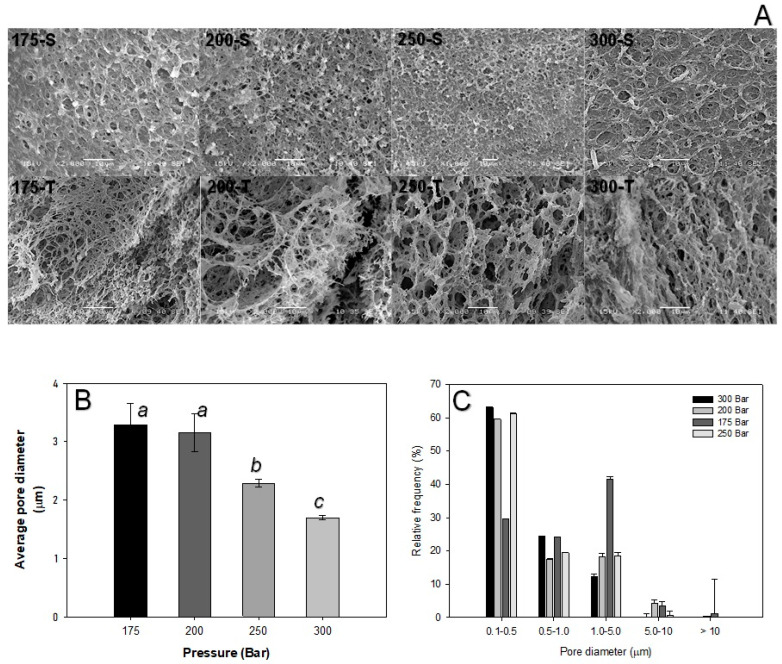
SEM micrographs of S_M_ after scCO_2_ (S: surface, T: transverse) (**A**), average diameter (**B**), and distribution of the pores (**C**) at different pressures and constant temperature (353 K). The different letters in the histogram mean that they are statistically different (Tukey–Kramer *p* ≤ 0.05).

**Figure 6 polymers-16-01569-f006:**
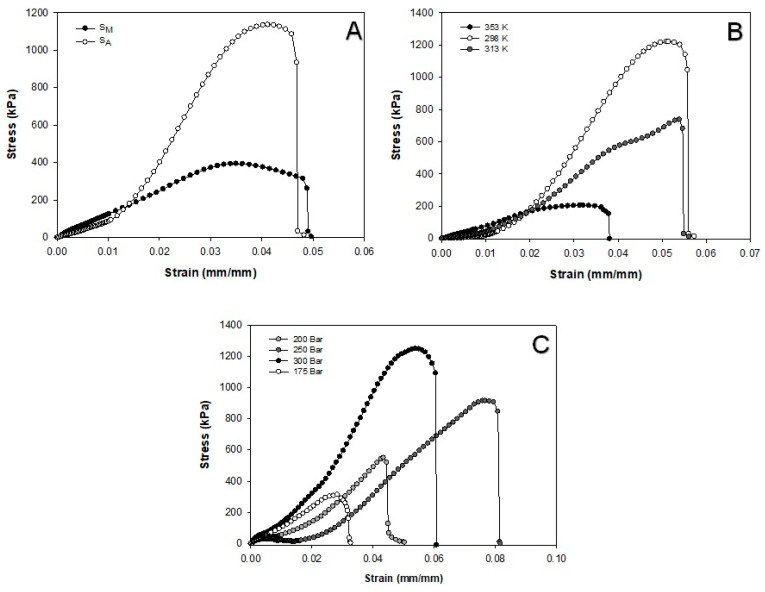
Stress–strain curves for SCF-treated S_A_ and S_M_ samples at 353 K and 175 bar (**A**) and for S_M_ samples at different temperature (**B**) and pressure (**C**) conditions.

**Figure 7 polymers-16-01569-f007:**
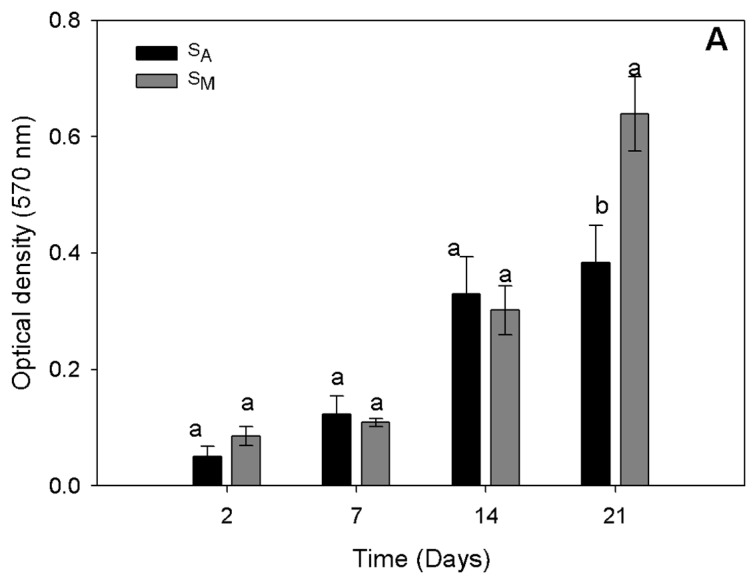
Proliferation of human osteoblasts cultured in scCO_2_-mediated porous chitin S_M_ and S_A_ scaffolds. Different letters in the histogram at the same days of cell culture mean that they were statistically different (Tukey–Kramer *p* ≤ 0.05).

**Figure 8 polymers-16-01569-f008:**
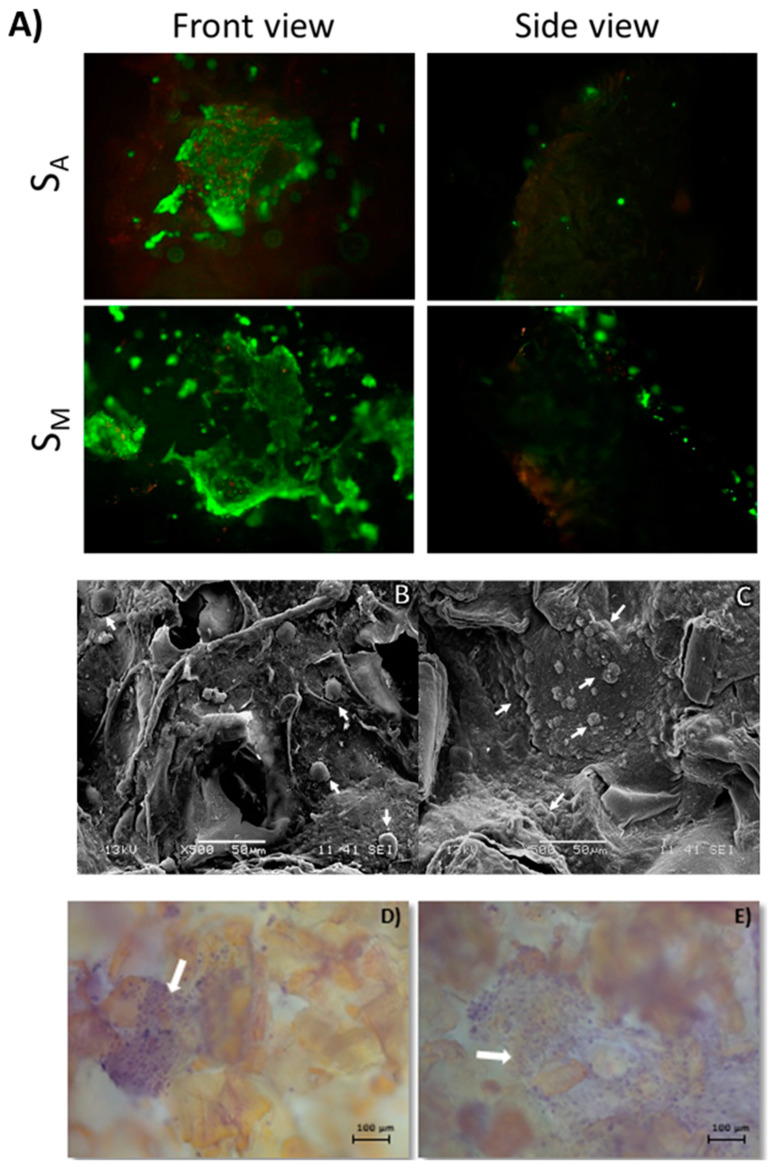
Cell viability of human osteoblasts cultured for 14 days in scCO_2_-mediated porous chitin S_M_ and S_A_ scaffolds. Shown from front (surface) and side (cell penetration) views (**A**). Viable cells are shown in green, while non-viable cells are shown in red. SEM micrographs of human osteoblasts (white arrows) cultured on scCO_2_-mediated porous chitin S_A_ (**B**) and S_M_ (**C**) scaffolds after 21 days of cell culture. Representative H&E staining micrographs showing the cell nuclei (white arrows) of human osteoblasts monolayers on scCO_2_-mediated porous chitin S_A_ (**D**) and S_M_ (**E**) scaffolds after 14 days of cell culture.

**Table 1 polymers-16-01569-t001:** Sorption of CO_2_ in S_M_ materials at several temperatures and pressures.

Pressure (bar)	Temperature (K)	Absorbed CO_2_ (mg)	Water Absorption Capacity (g H_2_O‧g scaffold^−1^)
175	298	2.6	2.2 ± 0.02 ^b^
	313	1.7	2.4 ± 0.2 ^b^
	353	16.6	3.0 ± 0.1 ^a^
175	353	18.5	3.5 ± 0.2 ^a^
200		2.3	2.6 ± 0.3 ^b^
250		1.4	2.0 ± 0.04 ^c^
300		1.6	2.1 ± 0.3 ^c^

Different letters in the columns mean that they are statistically different (Tukey–Kramer *p* ≤ 0.05).

**Table 2 polymers-16-01569-t002:** Physicochemical characteristics for S_A_ and S_M_ materials after the scCO_2_ foaming at 353 K and 175 bar.

Sample	C_I_ (%)	D_app_ (nm)	DA (%)	Contact Angle (θ)	Swelling (%)	Erosion (%)	Young’s Modulus (MPa)
S_A_	89.4 ± 0.5 ^a^	6.9 ± 0.1 ^a^	83.9 ± 0.2 ^b^	94.0 ± 7.61 ^a^	218.6 ± 2.7 ^b^	10.7 ± 0.9 ^b^	37.9 ± 8.2 ^a^
S_M_	76.8 ± 0.5 ^b^	5.2 ± 0.4 ^b^	84.4 ± 0.5 ^b^	31.9 ± 6.0 ^b^	326.7 ± 2.5 ^a^	22.4 ± 3.4 ^a^	14.4 ± 1.4 ^b^

Different letters in the rows mean that they are statistically different (Tukey–Kramer *p* ≤ 0.05).

**Table 3 polymers-16-01569-t003:** Physicochemical characteristics of S_M_ scaffolds upon different temperature and pressure in the scCO_2_ foaming.

Characteristic	Temperature (K) at 175 Bar			Pressure (bar) at 353 K			
	298	313	353	175	200	250	300
CI (%)	90.6 ± 0.9 ^a^	75.6 ± 0.4 ^b^	67.2 ± 0.2 ^c^	71.5 ± 1.2 ^c^	89.1 ± 1.2 ^a^	88.2 ± 0.4 ^a^	85.0 ± 0.8 ^b^
Dapp (nm)	6.9 ± 0.7 ^a^	5.6 ± 0.1 ^b^	4.8 ± 0.2 ^c^	4.6 ± 0.4 ^c^	6.7 ± 0.6 ^a^	6.5 ± 0.4 ^a^	6.2 ± 0.5 ^b^
DA (%)	90.5 ± 0.4 ^a^	88.2 ± 0.4 ^b^	84.8 ± 0.3 ^c^	77.7 ± 0.2 ^c^	87.8 ± 0.2 ^b^	89.4 ± 0.2 ^a^	90.2 ± 0.6 ^a^
Contact angle (θ)	52.5 ± 4.5 ^c^	79.91 ± 1.6 ^b^	11.8 ± 2.4 ^a^	37.5 ± 5.2 ^a^	46.8 ± 5.9 ^b^	56.2 ± 8.1 ^c^	55.1 ± 2.8 ^c^
Swelling (%)	302 ± 3.5 ^c^	324.7 ± 2.6 ^b^	377 ± 1.1 ^a^	355.1 ± 2.1 ^a^	302.2 ± 3.1 ^b^	245.8 ± 0.8 ^c^	208.4 ± 0.6 ^d^
Erosion (%)	14.2 ± 3.0 ^c^	17.4 ± 0.9 ^b^	20.6 ± 1.0 ^a^	28.1 ± 3.3 ^a^	25.5 ± 0.5 ^a^	20.8 ± 0.9 ^b^	19.5 ± 0.3 ^b^
Young’s modulus (MPa)	43.1 ± 3.6 ^c^	27.6 ± 4.7 ^b^	12.5 ± 1.6 ^a^	10.5 ± 3.0 ^c^	20.4 ± 5.3 ^b^	24.8 ± 6.9 ^b^	41.9 ± 10.8 ^a^

Different letters in the rows mean that they are statistically different (Tukey–Kramer *p* ≤ 0.05).

## Data Availability

Data are contained within the article and Appendix A.

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
