# Peer review of "Green Foaming of Biologically Extracted Chitin Hydrogels Using Supercritical Carbon Dioxide for Scaffolding of Human Osteoblasts"

_polymers, 2024, doi:10.3390/polym16111569_

Round 1

Reviewer 1 Report

Comments and Suggestions for Authors

In this manuscriptGreen foaming of biologically extracted chitin hydrogels using supercritical carbon dioxide for scaffolding of human osteo-blasts”,In the authors, methyl nanogels prepared with chitin hydrogel and methanol/calcium chloride solution were used as materials and treated with supercritical carbon dioxide (scCO2) to prepare porous scaffold materials for osteoblasts. Depending on the operating conditions and the addition of co-solvents, control of the morphology, porosity and physicochemical properties of the produced material is established. This results in a highly porous chitin scaffold. After the treatment of the hydrogel under supercritical conditions, the crystallinity index of chitin decreased significantly, and its swelling ability was significantly affected. However, there had some problems and questions, the comments were as followed:

1.        “The shrimp waste was thawed and mixed with sucrose (10 wt/wt%) and 24 h culture of Lactobacillus brevis in Man Rogosa Sharpe broth at 30°C as starter (5 vol/wt%).”Why were such parameters and cultures chosen?

2.        What does line 83 mean?

3.        Why use methanol as a co-solvent and what are the advantages compared to other inorganic solvents?

4.        The IR spectra were not seen, and the authors were advised to check and add the structural characterization at 3.2 of the IR spectra.

5.        Line 391”SM”instead”SM”.

6.        It is recommended that the author add a discussion section.

Author Response

Thank you very much for taking the time to review this manuscript. Please find the detailed responses below and the corresponding revisions/corrections highlighted in red in the re-submitted files

Point 1: In this manuscript “Green foaming of biologically extracted chitin hydrogels using supercritical carbon dioxide for scaffolding of human osteo-blasts”, In the authors, methyl nanogels prepared with chitin hydrogel and methanol/calcium chloride solution were used as materials and treated with supercritical carbon dioxide (scCO2) to prepare porous scaffold materials for osteoblasts. Depending on the operating conditions and the addition of co-solvents, control of the morphology, porosity and physicochemical properties of the produced material is established. This results in a highly porous chitin scaffold. After the treatment of the hydrogel under supercritical conditions, the crystallinity index of chitin decreased significantly, and its swelling ability was significantly affected. However, there had some problems and questions, the comments were as followed:

  1. The shrimp waste was thawed and mixed with sucrose (10 wt/wt%) and 24 h culture of Lactobacillus brevis in Man Rogosa Sharpe broth at 30°C as starter (5 vol/wt%).”Why were such parameters and cultures chosen?

Response 1: We acknowledge the reviewer opinion of the manuscript. We have included a brief explanation on the conditions of the bioprocess for chitin extraction as well as the supporting reference. All changes in the revised version of the manuscript are highlighted in red.

  1. What does line 83 mean?

Response 2: The subtitle (line 83) has been deleted.

  1. Why use methanol as a co-solvent and what are the advantages compared to other inorganic solvents?

Response 3: The methanol is a suitable organic solvent because it presents high miscibility in supercritical carbon dioxide, water is indeed an inorganic solvent, however it shows poor miscibility in this inorganic compressed fluid, and it also presents poor porogenic properties. Methanol is mostly reported as suitable porogenic co-solvent for foaming polymeric materials with scCO2, what is so called methanogels. Line 61 in the manuscript with the two references at the end of the sentence exemplifies that as follows: “Hydrogels and methanogels can be foamed by using scCO2 and co-solvents in a SCF system [6,10]” Additionally, the sentence in page 7clarifies the advantage of methanol as co-solvent and this as follows: “The methanol as co-solvent facilitated the solubility of CO2 in chitin, thus, favoring the absorption and interaction of this inorganic fluid with the polymer molecules by breaking intramolecular chain-bonding, and thereby modifying the crystalline arrangement.

  1. The IR spectra were not seen, and the authors were advised to check and add the structural characterization at 3.2 of the IR spectra.

Response 4: The structural characterization by assignation of the FTIR bands is thoroughly described in section 3.2 and appendix C shows the FT IR spectra.

  1. Line 391”SM”instead”SM”.

Response 5: The subscript M of SM has been corrected in the revised version of the manuscript

  1. It is recommended that the author add a discussion section.

Response 6: The Instructions for Authors in the Research Manuscript Sections explain that the Discussion may be combined with Results. Therefore, the Discussion section starts in line 234 as “Results and Discussions.”

Reviewer 2 Report

Comments and Suggestions for Authors

The authors have reported chitin-based hydrogels as potential scaffold for human osteoblasts by supercritical carbon dioxide processing. The morphology, porosity, and swelling of the hydrogels were studied under various conditions to better control of the scaffold’s characteristics, which eventually adequate adhesion and proliferation of osteoblasts. Overall the experiments seem to be carefully carried out and the writing is well organized, and the manuscript would be recommended for publication after considering a few points listed below. 

Line 136: NMR “spectrometer”, not “spectroscope”; what is the NMR solvent used? On what samples? Table 2 (and line 304) indicates that the NMR is acquired on gel materials after foaming, but how are they soluble for NMR determination?

It would be helpful to explain how surface and transverse SEM views (as in Fig. 4) were obtained.

Line 266: “the stretching of the carbonyl group were observed at 1085, 1125 and 1160 cm-1.” These peaks can be attributed to C-O single bonds, but certainly not the stretching of the carbonyl group. The same mistake also appears in Line 273-274.

Line 295: reference should be given for “a well-known pattern”.

Line 559: title needs to be specified.

Comments on the Quality of English Language

Language generally OK, but there are some minor grammatical and formatting errors that can be corrected upon careful reading. 

Author Response

Thank you very much for taking the time to review this manuscript. Please find the detailed responses below and the corresponding revisions/corrections highlighted in red in the re-submitted files.

The authors have reported chitin-based hydrogels as potential scaffold for human osteoblasts by supercritical carbon dioxide processing. The morphology, porosity, and swelling of the hydrogels were studied under various conditions to better control of the scaffold’s characteristics, which eventually adequate adhesion and proliferation of osteoblasts. Overall the experiments seem to be carefully carried out and the writing is well organized, and the manuscript would be recommended for publication after considering a few points listed below.

  1. Line 136: NMR “spectrometer”, not “spectroscope”; what is the NMR solvent used? On what samples?

Response 1: Spectrometer has been placed instead of spectroscope. DCl/D2O was used as solvent with 3-(trimethylsilyl) propionic acid as internal reference and added in the manuscript and it applies for all samples.

  1. Table 2 (and line 304) indicates that the NMR is acquired on gel materials after foaming, but how are they soluble for NMR determination?

Response 2:         All materials were soluble in DCl/D2O as described in materials and methods section.

  1. It would be helpful to explain how surface and transverse SEM views (as in Fig. 4) were obtained.

Response 3: "The samples were dried, and then coated with gold before microscopic examination. Transverse samples were also fractured and treated similarly." This text has been added in the methods section. 

  1. Line 266: “the stretching of the carbonyl group were observed at 1085, 1125 and 1160 cm-1.” These peaks can be attributed to C-O single bonds, but certainly not the stretching of the carbonyl group. The same mistake also appears in Line 273-274.

Response 4: It is correct carbonyl stretching is in the 1600-1800 cm-1 region, this has been corrected in the manuscript as 1085, 1125 and 1160 cm-1 bands correspond to C-O single bonds of the carbonyl groups in a modified sentence as follows: “The characteristic FTIR bands for the stretching of the C-O single bonds vibrations of the carbonyl groups were observed at 1085, 1125 and 1160 cm-1.” And in line 273 as follows “Other bands for the C-O are found at 1068 and 1153 cm-1

  1. Line 295: reference should be given for “a well-known pattern”.

Response 5: References 20 and 21 have been added at the end of that paragraph and marked in red in the revised manuscript.

  1. Line 559: title needs to be specified.

Response 6: The authors’ contributions are written according to the journal’s guidelines in the Instructions for Authors section, which do not require titles for the authors.

Round 2

Reviewer 1 Report

Comments and Suggestions for Authors

The revised manuscript can be accepted in the present form.